# Clinical Heterogeneity and Different Phenotypes in Patients with *SETD2* Variants: 18 New Patients and Review of the Literature

**DOI:** 10.3390/genes14061179

**Published:** 2023-05-29

**Authors:** Alejandro Parra, Rachel Rabin, John Pappas, Patricia Pascual, Mario Cazalla, Pedro Arias, Natalia Gallego-Zazo, Alfredo Santana, Ignacio Arroyo, Mercè Artigas, Harry Pachajoa, Yasemin Alanay, Ozlem Akgun-Dogan, Lyse Ruaud, Nathalie Couque, Jonathan Levy, Gloria Liliana Porras-Hurtado, Fernando Santos-Simarro, Maria Juliana Ballesta-Martinez, Encarna Guillén-Navarro, Hugo Muñoz-Hernández, Julián Nevado, Jair Tenorio-Castano, Pablo Lapunzina

**Affiliations:** 1CIBERER, Centro de Investigación Biomédica en Red de Enfermedades Raras, 28046 Madrid, Spain; alejandro.parra.externo@salud.madrid.org (A.P.); patripascual0@gmail.com (P.P.); palajara@gmail.com (P.A.); nataliagallegozazo@gmail.com (N.G.-Z.); guillen.encarna@gmail.com (E.G.-N.); jnevadobl@gmail.com (J.N.); jairantonio.tenorio@gmail.com (J.T.-C.); 2INGEMM-Idipaz, Institute of Medical and Molecular Genetics, 28046 Madrid, Spain; mario.cazalla16@gmail.com; 3ITHACA, European Reference Network, Hospital Universitario La Paz, 28046 Madrid, Spain; 4Clinical Genetic Services, Department of Pediatrics, NYU School of Medicine, New York, NY 10016, USA; rachel.rabin@nyulangone.org (R.R.); john.pappas@nyulangone.org (J.P.); 5Clinical Genetics, NYU Orthopedic Hospital, New York, NY 10010, USA; 6Clinical Genetics Unit, Complejo Hospitalario Universitario Insular-Materno Infantil de Las Palmas de Gran Canaria, 35016 Las Palmas de Gran Canaria, Spain; consultageneticasantana@gmail.com; 7Pediatrics Department, San Pedro de Alcántara Hospital, 10003 Cáceres, Spain; ignacio.arroyo@salud-juntaex.es; 8Genetics Unit, Hospital de Navarra, 31008 Pamplona, Spain; merce.artigas.lopez@navarra.es; 9Fundación Valle del Lili, Universidad Icesi, 760032 Cali, Colombia; hmpachajoa@icesi.edu.co; 10Division of Pediatric Genetics, Department of Pediatrics, Faculty of Medicine, Acibadem Mehmet Ali Aydinlar University, Istanbul 34752, Turkey; yasemin.alanay@acibadem.edu.tr (Y.A.); ozlem.dogan@acibadem.edu.tr (O.A.-D.); 11Rare Diseases and Orphan Drugs Application and Research Center (ACURARE), Acibadem Mehmet Ali Aydinlar University, Istanbul 34752, Turkey; 12Department of Genetics, APHP-Robert Debré University Hospital, 75019 Paris, France; lyse.ruaud@aphp.fr (L.R.); nathalie.couque@aphp.fr (N.C.); jonathan.levy@aphp.fr (J.L.); 13INSERM UMR1141, Neurodiderot, University of Paris Cité, 75019 Paris, France; 14Laboratoire de Biologie Médicale Multisites Seqoia-FMG2025, 75014 Paris, France; 15Línea de Investigación de Anomalías Congénitas y Enfermedades Huérfanas-Comfamiliar, Risaralda, Colombia; glolipo@utp.edu.co; 16Unidad de Diagnóstico Molecular y Genética Clínica, Hospital Universitario Son Espases, Idisba, 07120 Palma de Mallorca, Spain; fsantossimarro@gmail.com; 17Sección de Genética Médica, Hospital Clínico Universitario Virgen de la Arrixaca, 30120 Murcia, Spain; mjballesta@ucam.edu; 18Instituto Murciano de Investigación Biosanitaria (IMIB), 30120 Murcia, Spain; 19Department of Biology, Institute of Molecular Biology and Biophysics, ETH Zurich, 8092 Zurich, Switzerland; hugo.munoz@mol.biol.ethz.ch; 20Spanish OverGrowth Registry Initiative, La Paz University Hospital, 28046 Madrid, Spain; thesogriconsortium@gmail.com

**Keywords:** *SETD2*, Luscan–Lumish syndrome, Rabin–Pappas syndrome, intellectual developmental disorder, autosomal dominant 70, overgrowth, intellectual disability, autism spectrum disorder, MRD70, LLS, RAPAS

## Abstract

SETD2 belongs to the family of histone methyltransferase proteins and has been associated with three nosologically distinct entities with different clinical and molecular features: Luscan–Lumish syndrome (LLS), intellectual developmental disorder, autosomal dominant 70 (MRD70), and Rabin–Pappas syndrome (RAPAS). LLS [MIM #616831] is an overgrowth disorder with multisystem involvement including intellectual disability, speech delay, autism spectrum disorder (ASD), macrocephaly, tall stature, and motor delay. RAPAS [MIM #6201551] is a recently reported multisystemic disorder characterized by severely impaired global and intellectual development, hypotonia, feeding difficulties with failure to thrive, microcephaly, and dysmorphic facial features. Other neurologic findings may include seizures, hearing loss, ophthalmologic defects, and brain imaging abnormalities. There is variable involvement of other organ systems, including skeletal, genitourinary, cardiac, and potentially endocrine. Three patients who carried the missense variant p.Arg1740Gln in *SETD2* were reported with a moderately impaired intellectual disability, speech difficulties, and behavioral abnormalities. More variable findings included hypotonia and dysmorphic features. Due to the differences with the two previous phenotypes, this association was then named intellectual developmental disorder, autosomal dominant 70 [MIM 620157]. These three disorders seem to be allelic and are caused either by loss-of-function, gain-of-function, or missense variants in the *SETD2* gene. Here we describe 18 new patients with variants in *SETD2*, most of them with the LLS phenotype, and reviewed 33 additional patients with variants in *SETD2* that have been previously reported in the scientific literature. This article offers an expansion of the number of reported individuals with LLS and highlights the clinical features and the similarities and differences among the three phenotypes associated with *SETD2*.

## 1. Introduction

Overgrowth syndromes (OGS) comprise a heterogeneous group of disorders whose main characteristic is that either the weight, height, or head circumference, (often also occurring together) are above the 97th centile or 2–3 standard deviations (SD) above the mean for age, gender, and ethnic group [1]. Most of the OGS are associated with other clinical features that sometimes overlap between them, making the clinical diagnosis a challenge for both pediatricians and geneticists.

Luscan–Lumish syndrome (LLS) [MIM 616831] is an infrequent overgrowth disorder [2]. The main clinical features of this condition include macrocephaly, tall stature, intellectual disability, speech delay, autism spectrum disorder (ASD), and motor delay [3]. In 2020, Rabin et al. [4] described a series of 12 patients associated with a missense variant at codon 1740 of the *SETD2* gene. Patients mostly had microcephaly, intellectual disability, and multiple congenital abnormalities, such as congenital heart defects, abnormality of the skeletal system, and/or abnormality of the genitourinary system. The phenotypic association was named RAPAS [MIM 6201551], and all of these patients carried the same de novo missense variant, p.Arg1740Trp. The same authors also described three patients who carried another missense variant at the same amino acid position, p.Arg1740Gln in the *SETD2* gene. This variant was present in patients with a different phenotype compared to those with RAPAS, including mild global developmental delay, moderately impaired intellectual disability with speech difficulties, and behavioral abnormalities. More variable findings included hypotonia and dysmorphic features. This association was then named the intellectual developmental disorder, autosomal dominant 70 (MRD70) [MIM 620157]. The fact that these phenotypes were different to the classic LSS could be explained by a possible gain-of-function mechanism, or an effect in the epigenetic regulation of this gene [4].

LLS, RAPAS, and MRD70 are caused by heterozygous variants in the set domain-containing protein 2 (*SETD2)* gene located on chromosome 3p21.31. *SETD2* encodes a protein belonging to the methyltransferase family of proteins, which are involved in histone regulation, playing an important role in gene expression regulation [5]. SETD2 is also involved in other biological processes, such as DNA damage repair and DNA replication. Its main function is the trimethylation of lysine 36 on histone H3 (H3K36me3) [6]. Moreover, SETD2 methylates α-tubulin at lysine 40 during mitosis and cytokinesis, participating in the maintenance of genomic stability through its dual-function methyltransferase for chromatin and cytoskeleton [7]. Other genes belonging to the histone methyltransferase family have also been associated with overgrowth disorders (i.e., *DNMT3A* and *BRWD3*) [8,9].

As many other genes that are involved in overgrowth disorders, *SETD2* is associated with several neoplastic processes at the somatic level. SETD2 is absent or reduced in several cancers, supporting a tumor suppressive role of the protein [10]. In addition, somatic variants in *SETD2* have been found in many different cancers such as breast cancer, leukemia, and renal neoplasia [11,12].

Since the first detection of variants in *SETD2* as causative of ASD and a neurodevelopmental disorder [13,14], and the establishment of these variants as responsible for LLS, RAPAS, and MRD70 [2,4,15], only 33 patients have been reported to date, to the best of our knowledge. Most of these patients have been diagnosed by massive, paralleled sequencing technologies or NGS. Reported pathogenic or likely pathogenic variants in LLS, RAPAS and MRD70 comprise missense, nonsense, and frameshift variants in *SETD2*. In families in which segregation analysis was available, it was confirmed that most of the variants were de novo, and only in two patients was a vertical transmission reported [13].

Herein, we report 18 additional patients with variants in *SETD2* and a review of the clinical features found in LLS, RAPAS, and MDR70 patients from our cohort, and from all the individuals reported so far.

## 2. Material and Methods

### 2.1. Patients

Patients were selected from the Spanish Overgrowth Syndromes Registry Initiative (SOGRI), which comprises more than 2200 individuals and relatives with overgrowth disorders. This study was approved by the ethical committee of the Hospital Universitario La Paz (CEIm PI-446), and informed consent was obtained from all patients and/or their parents.

In addition to the SOGRI patients, a review of all previously reported patients in the scientific literature was made, and the phenotypes of these individuals were compared with those of the SOGRI described in this report. Additional patients were collected with collaborative support tools, including GeneMatcher [16].

### 2.2. Genetic Analysis

All patients from the Hospital Universitario La Paz were analyzed by a custom NGS panel using a Roche SeqCap EZ Kit (Roche, Basel, Switzerland) capture kit, and sequencing was performed with NextSeq500 technology (Illumina, CA, USA). A customized in-house bioinformatic pipeline was developed to analyze the raw data. This pipeline consisted of base calling, alignment, local realignment, duplicate removal, quality recalibration, data merging, variant detection, genotyping, and annotation. Quality control checkpoints were undertaken at numerous points to ensure the quality and integrity of the data, and as a result, the BAM and VCF files were obtained. Candidate variants that were obtained following a custom prioritization pipeline were validated by Sanger sequencing according to the standard procedures, and electropherograms were analyzed with Sequencher v4.1.4 (Genecodes, MI, USA). Patients from the other centers were analyzed either by their own NGS panels or with whole exome sequencing. Variant classification was made according to the qualitative American College of Medical Genetics and Genomics (ACMG) guidelines [17].

### 2.3. Protein Structural Analysis

The protein structure of wild type SETD2 1400–1800 and variants p.Glu1718Lys, p.Arg1740Trp, and p.Arg1740Gln were predicted using the Alphafold 2.1.1 neural network [18,19] and database of the scientific computing of ETH Zürich. The visualization was performed using UCSF ChimeraX [20,21].

## 3. Results

### 3.1. Molecular Results

We analyzed *SETD2* variants in 18 novel patients by NGS and identified 15 different genetic variants (four of them presented the same variant). Detailed information of the detected variants is shown in Table 1. None of the variants except for the p.Gln7Ter variant were previously reported in the literature. Nine out of the fifteen variants (60%) were absent in the pseudo-control population databases (gnomAD exomes, gnomAD genomes, Kaviar, 1000G, ESP, Beacon, and Bravo, respectively). The other six variants were found to have an extremely low population frequency: p.His866_Tyr871del: 0.00000657, p.Gln7Ter: 0.000244, p.Asp2100Gly: 0.00000657, and p.Asn1257Tyr: 0.00000657, respectively (data source: gnomAD genomes version 3.1.2). We also reviewed patients with *SETD2* variants previously reported in the literature, leading to 51 individuals being identified with variants in this gene reported so far. All *SETD2* variants reported are displayed in Figure 1 and listed in Table 1. To sum up, thirty-four genetic variants have been detected in this cohort of patients, comprising sixteen missense, nine frameshift, seven nonsense, one in-frame deletion, and one splicing variant, respectively. According to the guidelines of the American College of Medical Genetics (ACMG) [17], 15 variants were classified as pathogenic, 5 variants were classified as likely pathogenic, and the other 14 were classified as variants of unknown significance (VUS).

### 3.2. Clinical Features of Reported Patients

Clinical features of the patients reported herein are listed in Table 2, and pictures of several of these patients are shown in Figure 2. We also reviewed the clinical information of the 18 novel and those previously published patients with confirmed genetic variants in *SETD2* (n = 51). We separated these patients according to their phenotypes in three groups: Group 1—thirty-four patients with LLS; Group 2—fourteen patients with RAPAS, and Group 3—three patients with MRD70. Table 3 shows the frequency of clinical features in the three different groups of patients with variants in the *SETD2* gene. Figure 3 shows a distribution of the most relevant clinical features of each disorder. Intellectual disability was the most common clinical feature found among the three groups. Macrocephaly (67%), overgrowth (50%), and autism (50%) were identified as the clinical features with the highest frequency in patients with the LLS phenotype (group 1). Other common clinical features were speech delay (44.1%), developmental delay (38.2%), prominent forehead (32.4%), obesity (32.4%), motor delay (32.4%), and tall stature (29.4%). In patients with RAPAS, intellectual disability, microcephaly, abnormality of the skeletal system, absent speech, motor delay, developmental delay, and hypotonia were found in all patients. Hypertelorism, cerebellar hypoplasia, congenital heart defects, abnormality of the genitourinary system, failure to thrive in infancy, feeding difficulties, and micrognathia were observed in 12 patients (85.7%). Although the three patients with MRD70 share a few clinical features with patients of the group 2 (RAPAS phenotype), individuals with RAPAS are severely affected with multiple congenital anomalies and a profound intellectual disability. Mild intellectual disability (100%), abnormality of the skeletal system (66.6%), hypotonia (66.6%), and retrognathia (66.6%) were the most common clinical features identified in MRD70 individuals.

## 4. Discussion

*SETD2* encodes a lysine methyltransferase protein which trimethylates lysine 36 of histone H3 (H3K36me3) and methylates α-tubulin at lysine 40 [6,7]. Histone methylation is critical for embryonic development, and its dysregulation can lead to abnormalities in body patterning and defects in specific organ development. Loss of *SETD2,* which has previously been assessed through *SETD2* conditional knockouts in mice, revealed that this gene is essential for proper cortical arealization and corticothalamic projection formation. In addition, these *SETD2* knockout mice also displayed defects in social interaction, motor learning, and spatial memory, resembling LLS patients [27]. Moreover, knockout of *SETD2* results in defects in neuronal morphology transition, and therefore, in radial migration transition [28].

Here, we report 18 new patients with heterozygous variants in *SETD2*. So far, 33 patients have been reported with *SETD2* variants [2,3,4,13,15,22,23,24,25,26]. Thus, this report reviewed and summarized the information of 51 patients and emphasized the clinical heterogeneity in individuals carrying these *SETD2* variants. Pathogenic or likely pathogenic variants in *SETD2* can result in three different phenotypes: LLS, RAPAS, and MRD70, depending on the position of the variant in the protein [4]. The 51 patients we evaluated in this study were separated according to their phenotype: 34 LLS patients, 14 RAPAS patients, and 3 MRD70 patients.

Sixteen out of the eighteen novel patients described here had a clinical presentation compatible with LLS. The two remaining patients (patients #13 and #18) showed clinical features consistent with RAPAS. No patients with the MRD70 phenotype were found in our series. The results of the clinical study of our cohort supported the fact that macrocephaly, overgrowth, intellectual disability, autism, and delayed speech are the clinical features more commonly observed in LLS patients. Macrocephaly was the clinical feature with the highest frequency in LLS patients; it occurred in 23/34 patients, and 18 out of these 23 patients presented likely gene-disrupting variants (LGD), frameshift, or nonsense variants. Macrocephaly, together with overgrowth occurred in 15/34 (44.1%) patients. As is shown in Figure 2, LGD variants are not randomly distributed along the gene. It seems that there are two clusters of LGD variants, one between the codons 270 and 700 and the other one in the low charge region (LCR). Patients with LGD variants in these clusters tended to have more frequent macrocephaly, overgrowth, speech delay, autism, and developmental delays. In addition, a lack of LGD variants can be observed in the functional region of the protein (AWS-SET-PS domains). This may suggest that a highly deleterious effect on this region of the protein may produce an aberrant form incompatible with life development.

Patient 2 is a 13-years-old male with a head circumference >5SD. Gene panel sequencing enabled the detection of a heterozygous in-frame deletion variant c.2598_2615del (p.His866_Tyr871del) in *SETD2*. Genetic testing also detected a missense variant NM_000314.8:c.464A>G (p.Tyr155Cys) in *PTEN*. Both variants were inherited from the father, who presented with macrocephaly as well but with no other clinical features to resemble. Pathogenic variants in *PTEN* lead to the autosomal dominant disorder macrocephaly/autism syndrome [MIM #605309], among other overgrowth disorders and cancer processes at the somatic level. Patient 2 presented a very pronounced macrocephaly (>5SD). Both LLS due to *SETD2* and macrocephaly/autism syndrome due to *PTEN* pathogenic variants include macrocephaly as a common clinical feature. The head circumference measurement of this patient might be due to an additive effect of both genes. In fact, the additive effect of *PTEN* with other genes in several other malignancies has already been demonstrated [29].

Patients 3 to 6 and patient 20 all share the same nonsense variant NM_014159.7:c.19C>T (p.Gln7Ter) in *SETD2*. In patients 3 and 4, segregation analysis of the variant could not be performed, but vertical transmission of this variant was confirmed in patients 5 and 20. Both patients 5 and 20 inherited the variant from their mothers. The p.Gln7Ter variant results in a premature termination codon, which has been predicted to cause a truncation of the encoded protein, or the degradation of the transcript through the nonsense mediated decay (NMD) machinery. This variant is present in 40 alleles in gnomAD Exomes and gnomAD Genomes, with 39 of them belonging to the Latino subpopulation (with 0.000353 and 0.00229 allele frequencies in the Latino subpopulation in gnomAD Exomes and gnomAD Genomes, respectively). According to the guidelines of the ACMG, this variant has been classified as a variant of unknown significance (VUS). This variant is located in the first exon of the canonical transcript. However, for the rest of the transcripts, the variant is located within the 5′UTR region. Therefore, protein disruption could only take place in the canonical transcript. According to the GTex Portal, the canonical transcript is the second with the highest expression. Moreover, codon 12 of the canonical transcript is a methionine which, under the proper conditions, could act as a secondary translation initiator. Despite the fact that the five patients display several clinical features compatible with LLS, it seems that there is not enough evidence to classify the p.Gln7Ter variant as either pathogenic or likely pathogenic at this moment.

Patient 13 is a six year-old female who is heterozygous for the variant NM_014159.7: c.5152G>A (p.Glu1718Lys). She mainly presented with microcephaly, intellectual disability, developmental delay, motor delay, hypotonia, congenital heart defect, enlarged cisterna magna, and abnormality of the skeletal system (Table 2). She did not present clinical features common to other LLS patients. Despite the fact that she did not present the p.Arg1740Trp change that could point to RAPAS syndrome, microcephaly, intellectual disability, and abnormality of the skeletal system are clinical features present in all RAPAS patients. To date, the p.Arg1740Trp variant is the only one associated with RAPAS syndrome. The underlying mechanisms of this disorder are still unknown, though gain-of-function, effects on epigenetics regulation, or posttranslational modification of the cytoskeleton are putative suggested mechanisms [4]. Figure 4A,B shows a three-dimensional structure prediction of wild type SETD2. Under standard conditions, arginine 1740 is in an alpha helix, and interacts with the arginine 1744. At the same time, arginine 1744 has been predicted to be bonded to the glutamic acid 1718 by three hydrogen bonds. Figure 4C represents several three-dimensional structure predictions of SETD2 when the variants p.Arg1740Trp, p.Arg1740Gln, and p.Glu1718Lys occur. For the variant p.Arg1740Trp, the introduction of a nonpolar aromatic residue into an alpha helix may lead to a considerable structural alteration of the protein and thus affect its function. Therefore, the specific change at this position may lead to the development of RAPAS syndrome. Strikingly, in the same amino acid, there is another change (p.Arg1740Gln) which results in a missense substitution from the arginine amino acid to a glutamine residue. This change has been predicted to exhibit a minor effect on the protein function compared to the p.Arg1740Trp and may be also correlated with the differential phenotype observed in MRD70. Another option is that RAPAS and MRD70 are the same entity with highly heterogeneous clinical manifestations. Figure 4C(i) shows a comparison between the three-dimensional structure predictions of the wild type codon 1718 (Glu) and the changed one (Lys) in patient #13. Under standard conditions, wild type glutamic acid is predicted to be bonded to the arginine 1744 by three hydrogen bonds. When the c.5152G>A occurs, this Glu1718 is changed to a Lys and consequently, these three hydrogen bonds seem to disappear. In addition, glutamic acid is a negatively charged amino acid, while lysine is a positively charged amino acid. All this could lead to an effect in the structure or electronic environment of this region of the protein. As arginine 1744 is located in close proximity to arginine 1740, the missense variant p.Glu1718Lys may result in a similar alteration than the p.Arg1740Trp. This might explain why Patient 13 displays a similar phenotype to RAPAS patients.

Patient 18 is a four-year-old female with microcephaly, intellectual disability, hypotonia, ventriculomegaly, seizures, and abnormality of the skeletal system, among other clinical features. Her clinical presentation is compatible with RAPAS; however, similar to patient #13, she did not present the p.Arg1740Trp variant. In this patient, genetic testing revealed the missense variant NM_014159.7:c.6753C>G (p.Asp2251Glu). The functional interpretation of this finding remains inconclusive.

Although patient #13 and patient #18 have a consistent phenotype with RAPAS, they did not display the complete presentation of this syndrome, as neither of them had the characteristic brain abnormalities of RAPAS (cerebellar hypoplasia, hypoplasia of the pons, or hypoplasia of corpus callosum). To date, it seems that only patients with the variant p.Arg1740Trp in *SETD2* have the complete presentation of RAPAS.

In conclusion, we report 18 new patients with *SETD2* variants and review all the published patients to date raising a total of 51 patients described so far. Patients with *SETD2* variants are clinically heterogeneous and their clinical presentations seem to depend on the effect and/or the location of the variant among the protein. To date, pathogenic variants in *SETD2* are responsible for up to three different phenotypes. Loss-of-function variants located along almost the entire length of the gene lead to LLS, while missense variants at the specific position 1740 of the protein lead to at least two different phenotypes, named as RAPAS (p.Arg1740Trp) or MRD70 (p.Arg1740Gln). Strikingly, we report two patients with a change in different amino acid positions (p.Glu1718Lys and p.Asp2251Glu, respectively) with clinical presentations that are compatible with RAPAS, suggesting that other variants could lead to the same phenotype outside the amino acid position 1740. Our in silico protein model analysis revealed an interaction between amino acids 1744 and 1718, which can be associated with the distinctive phenotype in patients with variants at position 1740 of *SETD2*.

It is necessary to carry out further functional studies to understand the molecular mechanisms of these *SETD2* variants, and increase the number of patients assessed with variants in the *SETD2* gene to further define the phenotype splitting or lumping the nosology around this gene.

## Figures and Tables

**Figure 1 genes-14-01179-f001:**
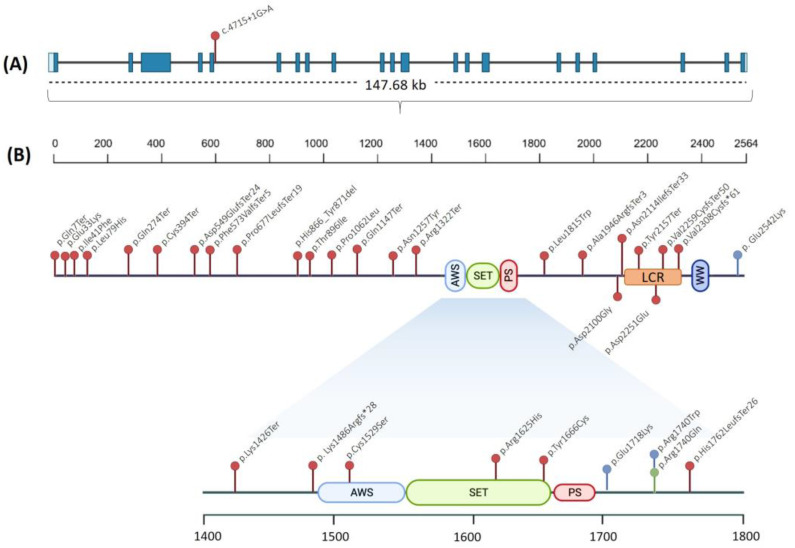
Variants identified in *SETD2*. (**A**) Exons and introns that conform to the *SETD2* gene according to the transcript number NM_014159.7. (**B**) The SETD2 protein, which is organized into different domains: AWS, associated with the SET domain; SET, Lysine N-methyltransferase domain; PS, post-SET domain; LCR, low charge region; and WW, WW domain. Red dots, variants associated with LLS; blue dots, variants associated with RAPAS; and green dots, variant associated with MRD70.

**Figure 2 genes-14-01179-f002:**
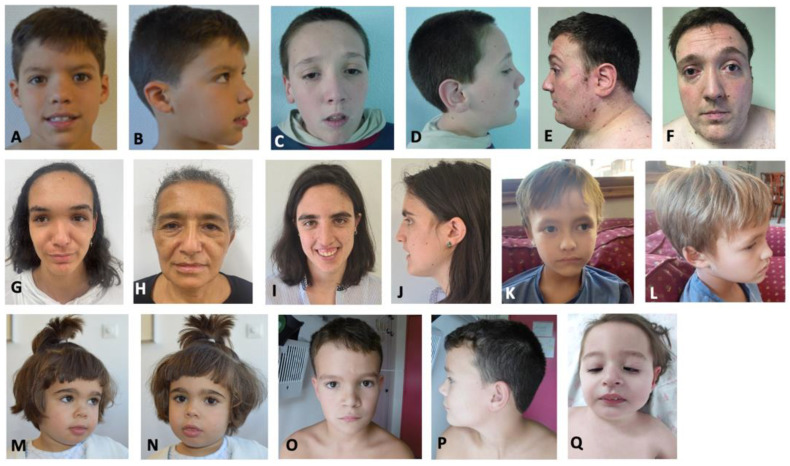
Facial dysmorphic features of several of the individuals reported herein. Patient 1 (**A**,**B**), patient 8 (**C**–**F**), patient 5 (**G**), patient 6 (**H**), patient 7 (**I**,**J**), patient 10 (**K**,**L**), patient 9 (**M**,**N**), patient 16 (**O**,**P**) and patient 18 (**Q**).

**Figure 3 genes-14-01179-f003:**
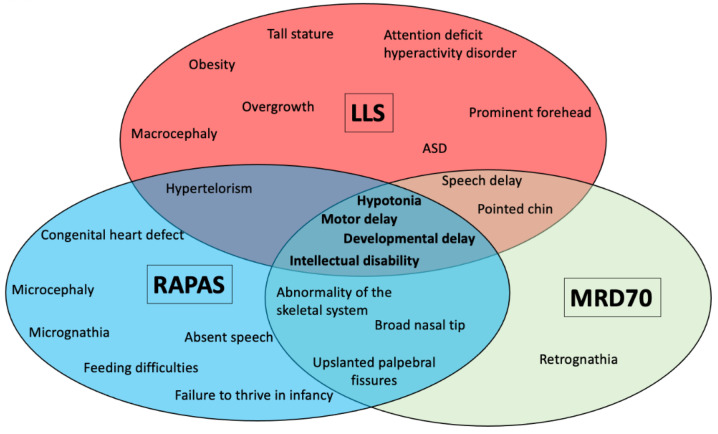
Venn diagram showing the most relevant clinical features of each *SETD2*- related disorder and their overlap between them. LLS, Luscan–Lumish syndrome; RAPAS, Rabin–Pappas syndrome; and MRD70, intellectual developmental disorder, autosomal dominant 70.

**Figure 4 genes-14-01179-f004:**
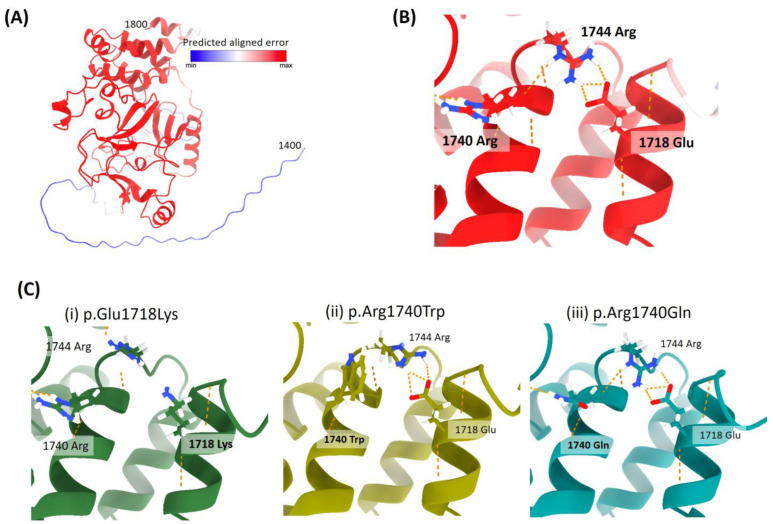
(**A**) Three-dimensional structure predictions of SETD2, from amino acid 1400 to 1800, based on NM_014159.7. Color key of predicted aligned error min = 20, ad max = 100. (**B**) Post_SET helical platform, showing the hydrogen bonds interactions occurring between the positions 1718-1740-1744 in purple. (**C**) Three-dimensional details of the variants (i) p.Glu1718Lys, (ii) p.Ar1740Trp, and (iii) p.Arg1740Gln, respectively.

**Table 1 genes-14-01179-t001:** Variants detected in the *SETD2* gene. Abbreviations: N/E, not evaluated; VUS, variant of uncertain significance; P, pathogenic; LP, likely pathogenic; LLS, Luscan–Lumish syndrome; RAPAS, Rabin–Pappas syndrome; and MRD70, intellectual developmental disorder, autosomal dominant 70. * Current age in years; ^†^ Allele frequency was estimated from several population pseudo-control databases: gnomAD genomes (v3.0), gnomAD exomes (v3.1), Kaviar (version 160204-Public), Beacon (v2.0), 1000 G, Phase III, and Bravo (TOVMed Freeze 8); ^§^ ACMG, American College of Medical Genetics.

Patient	Age *	Sex	Genomic Coordinate (hg38)	cDNA and Protein Location (NM_014159.7)	Exon/Intron	Mutation Type	Inheritance	Population Frequency ^†^	CADD Score	ACMG Prediction ^§^	Phenotype	Reference
1	14	Male	chr3:47116623	c.4586G>C (p.Cys1529Ser)	4	Missense	De novo	-	33	LP	LLS	This study
2	13	Male	chr3:47122021	c.2598_2615del (p.His866_Tyr871del)	3	In-frame deletion	Inherited from father	0.00000657	-	VUS	LLS	This study
3	17	Male	chr3:47163906	c.19C>T (p.Gln7Ter)	1	Nonsense	N/E	0.000244	35	VUS	LLS	This study
4	16	Male	chr3:47163906	c.19C>T (p.Gln7Ter)	1	Nonsense	N/E	0.000244	35	VUS	LLS	This study
5	21	Female	chr3:47163906	c.19C>T (p.Gln7Ter)	1	Nonsense	Inherited from mother	0.000244	35	VUS	LLS	This study
6	53	Female	chr3:47163906	c.19C>T (p.Gln7Ter)	1	Nonsense	N/E	0.000244	35	VUS	LLS	This study
7	26	Female	chr3:47056863	c.6921dupT (p.Val2308CysfsTer61)	15	Frameshift	N/E	-	-	P	LLS	This study
8	29	Male	chr3:47116749	c.4457_4460del (p.Lys1486ArgfsTer28)	4	Frameshift	N/E	-	-	P	LLS	This study
9	9	Female	chr3:47057485	c.6299A>G (p.Asp2100Gly)	15	Missense	N/E	0.00000657	28.1	VUS	LLS	This study
10	8	Male	chr3:47121949	c.2687C>T (p.Thr896Ile)	3	Missense	De novo	-	16.11	VUS	LLS	This study
11	4	Female	chr3:47122915	c.1717_1720del (p.Phe573ValfsTer5)	3	Frameshift	De novo	-	-	P	LLS	This study
12	5	Female	chr3:47121197	c.3439C>T (p.Gln1147Ter)	3	Nonsense	De novo	-	36	P	LLS	This study
13	6	Female	chr3:47088238	c.5152G>A (p.Glu1718Lys)	10	Missense	De novo	-	28.5	VUS	RAPAS	This study
14	15	Male	chr3:47120867	c.3769A>T (p.Asn1257Tyr)	3	Missense	Inherited from father	0.00000657	23.9	VUS	LLS	This study
15	13	Male	chr3:47120672	c.3964C>T (p.Arg1322Ter)	3	Nonsense	De novo	-	36	P	LLS	This study
16	11	Male	chr3:47124539	c.97G>A (p.Glu33Lys)	3	Missense	Inherited from mother	-	25.9	VUS	LLS	This study
17	13	Male	chr3:47017164	c.7624G>A (p.Glu2542Lys)	21	Missense	De novo	-	27.5	VUS	LLS	This study
18	4	Female	ch3:47057031	c.6753C>G (p.Asp2251Glu)	15	Missense	De novo	-	15.7	VUS	RAPAS	This study
19	15	Male	chr3:47123454	c.1182T>A (p.Cys394Ter)	3	Nonsense	Inherited from father	-	35	LP	LLS	[13]
20	15	Male	chr3:47163906	c.19C>T (p.Gln7Ter)	1	Nonsense	Inherited from mother	0.000244	35	VUS	LLS	[13]
21	26	Male	chr3:47124515	c.121A>T (p.Ile41Phe)	3	Missense	De novo	-	17.7	VUS	LLS	[13]
22	26	Female	chr3:47057443	c.6341delA (p.Asn2114IlefsTer33)	15	Frameshift	De novo	-	-	P	LLS	[13]
23	26	Female	chr3:47123816	c.820C>T (p.Gln274Ter)	3	Nonsense	N/E	-	34	P	LLS	[2]
24	29	Male	chr3:47084336	c.5444T>G (p.Leu1815Trp)	12	Missense	De novo	-	28	LP	LLS	[2]
25	24	Female	chr3:47122608	c.2028delT (p.Pro677LeufsTer19)	3	Frameshift	De novo	-	-	P	LLS	[15]
26	18	Male	chr3:47086306	c.5285_5286delAC (p.His1762LeufsTer26)	11	Frameshift	De novo	-	-	P	LLS	[22]
27	9	Male	chr3:47122969	c.1647_1667delinsAC (p.Asp549GlufsTer24)	3	Frameshift	De novo	-	-	P	LLS	[23]
28	27	Female	chr3:47057009	c.6775delG (p.Val2259CysfsTer50)	15	Frameshift	De novo	-	-	P	LLS	[23]
29	6	Female	chr3:47120360	c.4276A>T (p.Lys1426Ter)	3	Nonsense	N/E	-	37	P	LLS	[24]
30	8	Female	chr3:47103389	c.4874G>A (p.Arg1625His)	7	Missense	N/E	-	29	VUS	LLS	[24]
31	11	Male	chr3:47057313	c.6471T>A (p.Tyr2157Ter)	15	Nonsense	De novo	-	36	P	LLS	[24]
32	13	Male	chr3:47101476	c.4997A>G (p.Tyr1666Cys)	8	Missense	De novo	-	31	LP	LLS	[24]
33	22	Male	chr3:47124400	c.236T>A (p.Leu79His)	3	Missense	De novo	-	25.1	VUS	LLS	[25]
34	6	Male	chr3:47113875	c.4715+1G>A	intron 5	Splicing	De novo	-	34	P	LLS	[3]
35	6	Female	chr3:47121451	c.3185C>T (p.Pro1062Leu)	3	Missense	De novo	-	26.2	VUS	LLS	[3]
36–47	-	-	chr3:47088172	c.5218C>T (p.Arg1740Trp)	10	Missense	De novo	-	32	LP	RAPAS	[4]
48–50	-	-	chr3:47088171	c.5219G>A (p.Arg1740Gln)	10	Missense	De novo	-	28.5	VUS	MRD70	[4]
51	3	M	chr3: 47083945	c.5835_5836insAGAA (p. Ala1946ArgfsTer3)	12	Frameshift	De novo	-	-	P	LLS	[26]

**Table 2 genes-14-01179-t002:** Clinical features of reported patients. Detailed description of the clinical features at its frequency in the entire set of patients analyzed. Clinical features are standardized according to the human phenotype ontology (HPO).

Clinical Features		P1	P2	P3	P4	P5	P6	P7	P8	P9	P10	P11	P12	P13	P14	P15	P16	P17	P18	No. Patients	% Patients
HP:0000256	Macrocephaly																			11	61.1
HP:0001249	Intellectual disability																			9	50.0
HP:0001548	Overgrowth																			8	44.4
HP:0001263	Developmental delay																			8	44.4
HP:0001270	Motor delay																			8	44.4
HP:0011220	Prominent forehead																			7	38.9
HP:0000750	Speech delay																			7	38.9
HP:0000729	Autism spectrum disorder																			6	33.3
HP:0000337	Broad forehead																			6	33.3
HP:0001252	Hypotonia																			6	33.3
HP:0000348	Scoliosis																			5	27.8
HP:0001627	Congenital heart defect																			5	27.8
HP:0000483	Astigmatism																			4	22.2
HP:0000348	High forehead																			4	22.2
HP:0001513	Obesity																			4	22.2
HP:0000098	Tall stature																			4	22.2
HP:0000924	Abnormality of the skeletal system																			4	22.2
HP:0009890	High anterior hairline																			4	22.2
HP:0000739	Anxiety																			3	16.7
HP:0007018	Attention deficit hyperactivity disorder																			3	16.7
HP:0000708	Behavioral abnormality																			3	16.7
HP:0002007	Frontal bossing																			3	16.7
HP:0001999	Abnormal facial shape																			2	11.1
HP:0000077	Abnormality of the kidney																			2	11.1
HP:0001176	Long/Large hands																			2	11.1
HP:0003764	Nevus																			2	11.1
HP:0000486	Strabismus																			2	11.1
HP:0000316	Hypertelorism																			2	11.1
HP:0000252	Microcephaly																			2	11.1
HP:0001250	Seizure																			2	11.1
HP:0410263	Brain imaging abnormality																			2	11.1
HP:0000119	Abnormality of genitourinary system																			1	5.6
HP:0005616	Advanced bone age																			1	5.6
HP:0002119	Ventriculomegaly																			1	5.6
HP:0011427	Enlarged cisterna magna																			1	5.6
HP:0001601	Laryngomalacia																			1	5.6
HP:0010535	Sleep apnea																			1	5.6
HP:0000278	Retrognathia																			1	5.6
HP:0000464	Abnormality of the neck																			1	5.6
HP:0007763	Retinal telangiectasia																			1	5.6

**Table 3 genes-14-01179-t003:** Frequency of clinical features in the different groups of patients with variants in the *SETD2* gene. LLS, Luscan–Lumish syndrome; RAPAS, Rabin–Pappas syndrome; and MRD70, Intellectual developmental disorder, autosomal dominant 70.

HPO Terms Clinical Features		LLS Patients		RAPAS Patients		MRD70 Patients	
		No. Patients	% Patients	No. Patients	% Patients	No. Patients	% Patients
HP:0000256	Macrocephaly	23	67.6	0	0	0	0
HP:0001548	Overgrowth	17	50.0	0	0	0	0
HP:0000729	Autism spectrum disorder	17	50.0	0	0	0	0
HP:0001249	Intellectual disability	16	47.1	14	100	3	100
HP:0000750	Speech delay	15	44.1	0	0	3	100
HP:0001263	Developmental delay	13	38.2	14	100	3	100
HP:0011220	Prominent forehead	11	32.4	1	7.1	0	0
HP:0001513	Obesity	11	32.4	0	0	0	0
HP:0001270	Motor delay	11	32.4	14	100	2	66.7
HP:0000098	Tall stature	10	29.4	0	0	0	0
HP:0007018	Attention deficit hyperactivity disorder	9	26.5	0	0	0	0
HP:0000708	Behavioral abnormality	9	26.5	0	0	0	0
HP:0009890	High anterior hairline	8	23.5	1	7.1	0	0
HP:0000388	Otitis media	8	23.5	0	0	0	0
HP:0000337	Broad forehead	6	17.6	1	7.1	0	0
HP:0001833	Large feet	6	17.6	0	0.0	0	0
HP:0001252	Hypotonia	5	14.7	14	100	2	66.7
HP:0000483	Astigmatism	5	14.7	0	0	0	0
HP:0001176	Long/Large hands	5	14.7	0	0	0	0
HP:0000718	Aggressive behavior	5	14.7	0	0	0	0
HP:0000348	High forehead	5	14.7	0	0	0	0
HP:0000494	Downslanted palpebral fissures	5	14.7	0	0	0	0
HP:0000316	Hypertelorism	4	11.8	12	85.7	0	0
HP:0002007	Frontal bossing	4	11.8	1	7.1	0	0
HP:0000278	Scoliosis	4	11.8	8	57.1	0	0
HP:0000307	Pointed chin	4	11.8	0	0	1	33.3
HP:0001627	Congenital heart defect	4	11.8	12	85.7	0	0
HP:0000739	Anxiety	4	11.8	0	0	0	0
HP:0003764	Nevus	4	11.8	0	0	0	0
HP:0000276	Long face	4	11.8	0	0	0	0
HP:0002719	Recurrent infections	4	11.8	2	14.3	0	0
HP:0001250	Seizures	3	8.8	8	57.1	0	0
HP:0000924	Abnormality of the skeletal system	2	5.9	14	100	2	66.7
HP:0000272	Malar flattening	2	5.9	0	0	1	33.3
HP:0007360	Cerebellar hypoplasia	1	2.9	12	85.7	0	0
HP:0007370	Hypoplasia of the corpus callosum	1	2.9	9	64.3	0	0
HP:0000405	Conductive hearing impairment	1	2.9	7	50.0	0	0
HP:0001344	Absent speech	1	2.9	14	100	0	0
HP:0000252	Microcephaly	0	0	14	100	0	0
HP:0011968	Feeding difficulties	0	0	13	92.8	0	0
HP:0001531	Failure to thrive in infancy	0	0	12	85.7	0	0
HP:0000347	Micrognathia	0	0	12	85.7	0	0
HP:0000119	Abnormality of the genitourinary system	0	0	12	85.7	0	0
HP:0002553	Highly arched eyebrow	0	0	11	78.6	0	0
HP:0000455	Broad nasal tip	0	0	9	64.3	1	33.3
HP:0002791	Hypoventilation	0	0	9	64.3	0	0
HP:0009765	Low-hanging columella	0	0	9	64.3	0	0
HP:0000431	Wide nasal bridge	0	0	9	64.3	0	0
HP:0007763	Retinal telangiectasia	0	0	9	64.3	0	0
HP:0002902	Hyponatremia	0	0	8	57.1	0	0
HP:0000327	Hypoplasia of the maxilla	0	0	8	57.1	0	0
HP:0012110	Hypoplasia of the pons	0	0	8	57.1	0	0
HP:0000629	Periorbital fullness	0	0	8	57.1	0	0
HP:0007763	Retinal telangiectasia	0	0	8	57.1	0	0
HP:0012745	Short palpebral fissure	0	0	8	57.1	0	0
HP:0000582	Upslanted palpebral fissures	0	0	5	35.7	1	33.3
HP:0000278	Retrognathia	0	0	1	7.1	2	66.7

## Data Availability

Authors can confirm that all relevant data are included in the article.

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
