# Peer review of "Clinical Heterogeneity and Different Phenotypes in Patients with SETD2 Variants: 18 New Patients and Review of the Literature"

_genes, 2023, doi:10.3390/genes14061179_

Round 1

Reviewer 1 Report

The paper "Clinical heterogeneity and different phenotypes in patients with SETD2 variants: 18 new patients and review of the literature" is a complete review on SETD2-related disorders, which gets also new insight thanks a novel case-series. The article is well presented, and  text is broken out per sections according to the journal required style. I've only few minor comments as follow:

- table 2 (very hard to read): I'd move on supplementary and/or I'd suggest to turn these data into a heatmap image; 

- I will strengthen in discussion the concept of uniqueness of genotype-phenotype correlation for each SETD2-related disorder, otherwise it could be proposed the existence of an SETD2-(endo)phenotypes (for example of MRD70 in RAPAS) also i.e. by evaluating the variants' segregation domains which appears quite ubiquitous only for the LLS, and very specific for the remaining forms. 

none

Reviewer 2 Report

Review of the manuscript

Clinical heterogeneity and different phenotypes in patients with SETD2 variants: 18 new patients and review of literature by Parra et al.

This is an important addition to the databases on genetic variants and phenotype variability of the histone methyltransferase gene SETD2 written by a large international consortium of clinicians and researchers. The heterogeneity of the clinical phenotypes and the nosologic problem of their “splitting and lumping” is acknowledged by the authors.

There is one paper that was published in march 2023 with a case report and a review of published cases. It includes also 3D-protein modelling of the variant of the patient. This case should be added to the list of patients and to references of the manuscript.

Wu Y, Liu F, Wan R, Jiao B. A novel SETD2 variant causing global development delay without overgrowth in a Chinese 3-year-old boy. Front. Genet., 14; 2023,
https://doi.org/10.3389/fgene.2023.1153284

In addition to the discussion relating to the effects of protein structure of the variants, I would have wished to have some information on what is known from isolated cells and from animal models.  There are some recent papers to suggest to be referred:

Cells:
Shen, T., Ji, F., Wang, Y., Lei, X., Zhang, D., and Jiao, J. (2018). Brain-specific deletion of histone variant H2A.z results in cortical neurogenesis defects and neurodevelopmental disorder. Nucleic Acids Res. 46, 2290–2307. doi: 10.1093/nar/gkx1295

Animal models (SETD2 knock-out and conditional knock-out experiments):
Xie, X., Wang, S., Li, M., Diao, L., Pan, X., Chen, J., et al. (2021). alpha-TubK40me3 is required for neuronal polarization and migration by promoting microtubule formation. Nat. Commun. 12:4113. doi: 10.1038/s41467-021-24376-2

Xu, L., Zheng, Y., Li, X., Wang, A., Huo, D., Li, Q., et al. (2021). Abnormal neocortex arealization and Sotos-like syndrome-associated behavior in Setd2 mutant mice. Sci. Adv. 7:eaba1180. doi: 10.1126/sciadv.aba1180

Small comments to the authors:
1.    Line 38: Omit “recently”
2.    line 49: Omit “also”
3.    line 49: Isn’t  “mild global developmental delay” included in the “moderately impaired intellectual disability” or what is the difference?
4.    line 118: “revision” or review?  Or reevaluation?
5.    line 123: “our Center”, please specify which one, this is a multi-center study.
6.     Figure 1. It’s a little difficult for the reader: the figure A is above B whereas in the legend there is B before A. I suggest that the order in the figure is changed so that the whole gene is up there with exons and introns and the bp bar, then underneath there would be the protein structure and then the magnified AWS+SET+PS domains area.
7.    Table 2 is a little confusing with a seemingly random order of the 18 novel patient cases. Especially if the symptom listed is reported in just one patient, I think this information ie. this part of the table, could be omitted for practical reasons. Was the clinical evaluation carried out in a similar manner?
8.    Table 3, lines 214-215 and lines 219-222: It is striking that only 23 of the 33 LLS patients had macrocephaly and only 15 had macrocephaly and overgrowth.  So, how can you call those without macrocephaly Luscan-Lumish syndrome? Would it be better in future to create  another “umbrella” name for all the phenotypes that have been described? SETD2-syndrome? This is an ontological problem.
9.    Lines 230-240: Could the macrocephaly of patient 2 be caused just by PTEN variant?  Is it then another syndrome?
10.    Figure 4. Protein modelling techniques were also based on the Alphafold Database by Wu et al (2023), but images differ. They used Pymol. I hope that the figures B and C in the manuscript would be bigger or with more evident colors.  It is a demanding task to recognize the differences and understand their effects.
11.    Line 331: Are the “resources” (the 15 names + SOGRI) all clinicians contributing with new cases? Back to point 7, were there rules how to examine the patients?
12.    As stated in my general opinion, I suggest adding a couple of sentences on what experiments of isolated nerve cells and animal models suggest of the role / roles played by this gene and its variants. 
